# Proteomic Time-Course Analysis of the Filamentous Anoxygenic Phototrophic Bacterium, *Chloroflexus aurantiacus*, during the Transition from Respiration to Phototrophy

**DOI:** 10.3390/microorganisms10071288

**Published:** 2022-06-25

**Authors:** Shigeru Kawai, Shigeru Shimamura, Yasuhiro Shimane, Yusuke Tsukatani

**Affiliations:** 1Institute for Extra-Cutting-Edge Science and Technology Avant-Garde Research (X-star), Japan Agency for Marine-Earth Science and Technology (JAMSTEC), Yokosuka 237-0061, Japan; shimas@jamstec.go.jp (S.S.); yshimane@jamstec.go.jp (Y.S.); 2Department of Biological Sciences, Tokyo Metropolitan University, Hachioji 192-0397, Japan; 3Research Institute of Industrial Technology, Toyo University, Kawagoe 350-8585, Japan

**Keywords:** *Chloroflexus aurantiacus*, proteomic analysis, photosynthesis, respiration, bacteriochlorophyll, chlorosome, alternative complex III

## Abstract

*Chloroflexus aurantiacus* is a filamentous anoxygenic phototrophic bacterium that grows chemotrophically under oxic conditions and phototrophically under anoxic conditions. Because photosynthesis-related genes are scattered without any gene clusters in the genome, it is still unclear how this bacterium regulates protein expression in response to environmental changes. In this study, we performed a proteomic time-course analysis of how *C. aurantiacus* expresses proteins to acclimate to environmental changes, namely the transition from chemoheterotrophic respiratory to photoheterotrophic growth mode. Proteomic analysis detected a total of 2520 proteins out of 3934 coding sequences in the *C. aurantiacus* genome from samples collected at 13 time points. Almost all proteins for reaction centers, light-harvesting chlorosomes, and carbon fixation pathways were successfully detected during the growing phases in which optical densities and relative bacteriochlorophyll *c* contents increased simultaneously. Combination of proteomics and pigment analysis suggests that the self-aggregation of bacteriochlorophyllide *c* could precede the esterification of the hydrophobic farnesyl tail in cells. Cytoplasmic subunits of alternative complex III were interchanged between oxic and anoxic conditions, although membrane-bound subunits were used for both conditions. These data highlight the protein expression dynamics of phototrophy-related genes during the transition from respiration to phototrophy.

## 1. Introduction

*Chloroflexus aurantiacus* is a thermophilic filamentous anoxygenic phototrophic bacterium in the phylum *Chloroflexota* [1]. This bacterium grows chemoheterotrophically under oxic dark conditions and photoheterotrophically under anoxic light conditions [2]. *Chloroflexus* species and the closely related *Roseiflexus* species are widely distributed in microbial mats in hot springs, where thermophilic cyanobacteria usually coexist [3,4]. Because cyanobacterial oxygenic photosynthesis causes drastic changes in O_2_ concentration over a diel cycle, co-existing microbes need to switch their metabolisms for adaptation to surrounding environments [5,6,7,8]. Therefore, the regulatory systems of protein expression are essential for the growth of *Chloroflexus* species in natural environments.

The photosynthetic apparatuses of *C. aurantiacus* consist of type II photochemical reaction center (RC) complexes, peripheral membrane-bound light-harvesting complexes, and organelle-like light-harvesting vesicles called chlorosomes [9]. The type II RC of *C. aurantiacus* is composed of core PufLM subunits, which are bound to bacteriochlorophyll (BChl) *a*, and a cytochrome *c* subunit, PufC, containing four hemes. A cryo-EM study using the RC from *Roseiflexus castenholzii* revealed that the RC core complex structure is similar to those of purple bacterial RCs [10]. Chlorosomes are composed of self-aggregated BChl *c* surrounded by a lipid monolayer envelope. A long axis of the chlorosome vesicle forms a crystalline-like structure, called a baseplate, consisting of a large number of CsmA proteins. The lipid monolayer envelope of chlorosomes also contains several other proteins, e.g., CsmMNPOY in the case of *C. aurantiacus*, CsmBCDEFHIJX in *Chlorobaculum tepidum* (*Chlorobi*), and CsmVRSTU in *Chloracidobacterium thermophilum* (*Acidobacteria*) [11,12,13]. It is noted that CsmA is the sole common chlorosome protein that shows sequence similarities among the three phototrophic phyla. It has been unclear how BChl *c* self-aggregates are packed inside chlorosome envelopes. Phototrophic members in *Chlorobi* and *Acidobacteria* possess the BChl *a*-containing antenna protein, called the FMO protein, which is located between chlorosomes and RC complexes, but *C. aurantiacus* does not have the FMO protein [14,15,16,17]. Instead, *C. aurantiacus* utilizes peripheral antenna proteins, called LH complexes, which attach to RC complexes in membranes. The LH complex is composed of two transmembrane helices, α- and β-peptides, which are encoded by the *pufA* and *pufB* genes, respectively.

BChl pigments are synthesized through multistep enzymatic reactions in cells of phototrophic micro-organisms [18,19]. However, the coexistence of chlorophyllous pigments and O_2_ causes the generation of reactive oxygen species. Therefore, in the case of purple bacteria (*Proteobacteria*) genes involved in BChl biosynthesis (*bch*) as well as in carotenoid biosynthesis (*crt*) and RC complexes (*puf*) are clustered together in a single location in the genome for the control of gene transcription under certain environments. The location in the genome is called a photosynthetic gene cluster (PGC), and gene expression in a PGC is suppressed under oxic conditions in purple bacteria [20,21]. In contrast, genome analysis of *C. aurantiacus* revealed that photosynthesis-related genes demonstrate scattered distribution in the genome of the bacterium [22]. It is not yet known how *C. aurantiacus* regulates the expression of photosynthesis-related genes in response to environmental changes or even whether the expression of the scattered genes is controlled concertedly.

Alternative complex III (ACIII) was found as a substitute for cytochrome *bc*_1_ complex in *C. aurantiacus* [23] and in the nonphotosynthetic bacterium *Rhodothermus marinus* [24,25]. Thus far, ACIII has been found in many other bacteria that do not possess cytochrome *bc*_1_ complex [26]. While nonphotosynthetic bacteria such as *R. marinus* use ACIII in the respiratory electron transfer chain, *C. aurantiacus* utilizes ACIII in both the respiratory and photosynthetic electron transfer chains [27]. In *Chloroflexus* species, two paralog gene sets for ACIII have been found, and it has been proposed that *C. aurantiacus* uses a protein complex called ACIII-C_p_ for photosynthetic electron transport and another protein complex, called ACIII-C_r_, for the respiratory chain [22,28,29].

In this study, we performed a proteomic time-course analysis of *C. aurantiacus* in the transition from respiration to phototrophy to investigate the expression dynamics of proteins involved in the biogenesis of the photosynthetic apparatus, electron transport chains, and central carbon metabolisms. Moreover, we analyzed the growth curve, absorption spectra, and pigment compositions of the bacterium during cultivation.

## 2. Materials and Methods

### 2.1. Bacterial Growth Conditions

*C. aurantiacus* strain J-10-fl^T^ (=DSM635) was precultured heterotrophically under oxic conditions in the dark at 48 °C with vigorous shaking in AY medium [30]. Ten milliliters of preculture were transferred to 5 L of fresh AY medium equipped in a 10 L thermostat fermenter. The oxic dark cultivation in the fermenter was performed by stirring at 350 rpm with air bubbling at 48 °C for 2 days. The fermenter was then illuminated by incandescent light with irradiance of 20 µmol photons s^−1^ m^−2^ from the front. Sodium acetate (7.5 g per 5 L) was aseptically added as the nutrient for photoheterotrophic growth. In the preliminary experiment, oxygen in the headspace was replaced by N_2_ to achieve strict anoxic conditions, but no growth was observed for at least 30 days. Therefore, 2% O_2_ (*v*/*v*) was supplemented after the N_2_ flush, after which the photoheterotrophic cultivation was started with stirring at 50 rpm at 48 °C. Absorption spectra of phototrophic cultures were measured using a UV-2600 spectrophotometer (Shimadzu, Kyoto, Japan). Cells were harvested by centrifugation on 13 days during 39 days of light cultivation. The cells harvested for proteome analysis and pigment analysis were washed three times with 100 mM triethylammonium bicarbonate (TEAB) buffer (pH 8.6) and 5 mM phosphate buffer (pH 7.5), respectively. The cell pellets were stored at −30 °C until use.

### 2.2. Protein Sample Preparations for LC-MS Proteomic Analysis

Cell pellets were suspended in 300 µL of 100 mM TEAB buffer containing 2 mM PMSF. The suspended cells were disrupted by sonication for 6 min (30-s sonication periods with 30-s intervals) using a Qsonica 700 sonicator (20 kHz frequency) equipped with a Qsonica 431C2 Cup Horn and a Qsonica Compact Recirculating Chiller (Qsonica, Newtown, CT, USA). The protein concentrations of the solution were determined using a Qubit Protein assay kit (Thermo Fisher Scientific, Waltham, MA, USA). The cell extracts corresponding to 10 µg of proteins were evaporated to dryness, and the dried proteins were dissolved in the denaturing buffer consisting of MPEX PTS Reagent Solutions A and B (GL Science, Tokyo, Japan). The denatured proteins were reduced by the addition of dithiothreitol (MS grade, Thermo Fisher Scientific) at a final concentration of 25 mM. The solution was incubated at 95 °C for 5 min and then at room temperature for 25 min. Iodoacetamide (MS grade, Thermo Fisher Scientific) was then added at a final concentration of 25 mM, and the solutions were incubated at room temperature for 30 min for alkylation. Protein digestion was initiated by the addition of 100 ng trypsin protease (MS grade, Thermo Fisher Scientific), and the solution was incubated at 37 °C for 3 h. Subsequently, 100 ng of trypsin protease was added again, and the proteins were further digested at 30 °C overnight. Detergents in the samples were removed using MPEX PTS Reagent Solutions C and D (GL Science) according to the manufacturer’s instructions. Residual surfactants were removed using Pierce Detergent Removal Spin Column (Thermo Fisher Scientific). The digested protein solutions were evaporated to dryness. The dried proteins were resuspended in 2% acetonitrile and 0.1% trifluoroacetic acid in water and then subjected to the LC-MS/MS analysis as described below.

Peptides were separated using an Ultimate 3000 RSLCnano system (Thermo Fisher Scientific) with a reverse-phase Zaplous alpha Pep-C18 column (3 μm, 120 Å, 0.1 × 150 mm; AMR, Tokyo, Japan). The gradient was composed of solvent A (0.1% formic acid in water) and solvent B (100% acetonitrile). At the time of injection, the mobile phase was 5% solvent B at a flow rate of 500 nL min^−1^. Solvent B was linearly increased to 45% over 100 min. The column temperature was set at 35 °C via a nano-electrospray ion source (Dream Spray, AMR, Tokyo, Japan). The Orbitrap Fusion Tribrid mass spectrometer (Thermo Fisher Scientific) was operated in the positive ion mode. For peptide ionization, an electrospray voltage of 1.7 kV was applied, and the ion transfer tube temperature was set at 250 °C. All MS spectra were acquired in the Orbitrap mass analyzer (*m*/*z* range: 350–1800, resolution: 120,000 FWHM) with EASY-IC internal mass calibration. MS/MS spectra resulting from collision-induced dissociation (CID) fragmentation were acquired in an ion trap mass analyzer (*m*/*z* range: auto, scan rate: rapid).

### 2.3. Data Analysis

Protein identification, label-free quantification of detected protein derivatives, and principal component analysis (PCA) were performed with the Proteome Discoverer ver. 2.2 software package (Thermo Fisher Scientific). The search for acquired spectra was carried out using the SEQUEST HT search algorithm against a list of CDSs identified in the genome of *C. aurantiacus*. The following search parameters were used: trypsin was specified as protease allowing two missed cleavage sites at the maximum; cysteine carbamidomethylation was set as a fixed modification; methionine oxidation was set as a variable modification; and maximum error tolerances in MS and MS/MS were set at 10 ppm and 0.6 Da, respectively. Peptides corresponding to a <1% protein false discovery rate (FDR) were used in the calculations.

Relative values of peptide detection for each protein are calculated for the comparison among 13 sampling dates. The calculated relative abundance was normalized with a percentage in each protein. The Sum PEP score of each protein was calculated in Proteome Discoverer ver. 2.2.

### 2.4. Pigment Extraction and HPLC Analysis

Pigments were extracted from the harvested cells with acetone–methanol (7:2, *v*/*v*). The solvent was evaporated to dryness by a stream of N_2_ gas, and the dried pigments were dissolved in the HPLC solvent described below. The pigment solution was filtered using a Cosmonice filter (0.45 µm; Nacalai Tesque, Kyoto, Japan); 0.1 mM of ammonium acetate was added to the filtrated solution before injection. Reverse-phase HPLC measurements were performed using a Cosmosil 5C18-AR-II (5 µm, 4.6 × 250 mm, Nacalai Tesque) column and the mobile phase of solvent A (methanol:acetonitrile:water, 42:33:25, *v*:*v*:*v*) and solvent B (methanol:acetonitrile:ethyl acetate, 50:20:30, *v*:*v*:*v*). The elution was conducted by the following method as previously described, with a modification [31,32]. The initial composition of solvents was 30% solvent B. The linear gradient was increased to 100% solvent B for 52 min followed by 6 min of constant flow. The solvent composition was then gradually returned to 30% solvent B over 2 min and held for 5 min.

## 3. Results

### 3.1. Growth Profiles

Growth curves of *C. aurantiacus* after the illumination started are shown in Figure 1. The growth profiles were evaluated using two sets of data: cell densities by measuring absorbance at 600 nm (A_600_) (Figure 1A) and relative BChl *c* contents (A_740_/A_600_) (Figure 1B). The A_740_ value is reflected from an absorption band from BChl *c* aggregates in chlorosomes. During the first 6 days, the A_600_ value decreased from 0.36 to 0.10 and relative BChl *c* contents did not change. The period of days 0 to 5 was designated the first lag phase (Figure 1, orange). During the next period, from days 6 to 10, the relative BChl *c* contents increased slightly while cell density values did not change. This period was designated the second lag phase (Figure 1, yellow). Significant increases in both cell densities and relative BChl *c* contents were observed from days 12 to 17 (designated as the first growing phase; Figure 1, light green). The doubling time during this period was 3.2 ± 0.5 days with the use of A_600_ values measured at days 12 to 17. After day 17, no increase in relative BChl *c* contents was observed, but cell densities in the A_600_ values kept increasing, with a doubling time of 11.8 ± 1.6 days (calculated by A_600_ values measured at days 20 to 32). This period was designated the second growing phase (Figure 1, dark green).

### 3.2. Protein Expression of Photosynthetic Proteins in C. aurantiacus

Proteomic analysis of *C. aurantiacus* in the transition from respiration to phototrophy was performed using protein samples collected at 13 different time points after illumination (days 0–39). Peptides derived from a total of 2520 proteins out of 3934 CDSs in deduced genome sequences of *C. aurantiacus* were successfully detected from the LC-MS/MS proteomic analysis of 2 duplicates of the 13 sampling dates. All proteomic datasets were compared using PCA (Appendix A). The PCA data showed that duplicates derived from the same sampling dates were clustered together, confirming the reproducibility of proteome data among duplicates.

Figure 2 shows the expression patterns of protein subunits and enzymes related to RC complexes, chlorosomes, and bacteriochlorophyll and carotenoid biosynthesis. Relative values of peptide detection for each protein among 13 sampling dates are shown in Figure 2 and Figure 3 as percentages indicating abundance, where “100%” indicates the highest protein expression among all sampling dates. Sum PEP scores indicate the relative values of protein amounts for the comparison of protein expression levels between different CDSs. Almost all the photosynthetic proteins were successfully detected with high abundance in the growing phase (days 12–39) compared to the lag phase (days 0–10), except PufA, BchX, BchG, CrtB, and CrtY (Figure 2). The abundance of peptides for RC complexes and chlorosomes increased along with cellular growth, indicating that these proteins support phototrophic growth. Peptides derived from PufLMC subunits of the RC and all of the Csm proteins were detected during the first and second growing phases, with the highest abundance at the last date of sampling (Figure 1, day 39). On the other hand, peptides from Bch enzymes for BChls *a* and *c* biosynthesis were already detected in the second lag phase, and several Bch enzymes became the most abundant on day 12 or 14 of the first growing phase (BchHDEYZCU, BciBC, and AcsF). The expression patterns of Bch proteins were consistent with the observation of a slight increase in relative BChl *c* content in the second lag phase and rapid increase in the first growing phase (Figure 1B). These results suggest that BChl biosynthesis precedes the formation of the RC-LH complex and chlorosomes.

In contrast, some enzymes for carotenoid biosynthesis (CrtO and CrtI-1) were expressed in the lag phase (Figure 2, days 0–10). CrtO is carotenoid ketolase, and ketolated carotenoids are known to scavenge reactive oxygen species [32,33]. The expression of CrtO ketolase was likely to be finished at around days 12–14, concomitant with the initiation of the expression of RCs and chlorosomes. This implies that oxygen added to the fermenter was consumed for up to 10–12 days, which may have initiated the phototrophic growth mode.

### 3.3. Photosynthetic and Respiratory Electron Transfer Chain

ACIII of the photosynthesis version in *C. aurantiacus* comprises seven subunits: penta-heme subunit ActA, iron-sulfur binding subunit ActB, transmembrane subunits ActCDF, mono-heme subunit ActE, and membrane-bound subunit ActG [29]. *C. aurantiacus* has all of the seven genes for ACIII-C_p_ in a single operon (Caur_0621–0627). ActABEG (C_p_) demonstrated the highest abundance in the second growing phase, whereas peptides derived from ActCDF (C_p_) already reached the highest abundance in the second lag phase following relatively high abundance (30~80%) during the first and second growing phases (Figure 3). Peptides derived from auracyanin A and auracyanin B, which are copper-containing soluble electron carrier proteins assumed to be substitutes of soluble cytochrome *c*, were also detected. The expression of auracyanin A was relatively stable during the whole period, with the lowest abundance being 36%. In contrast, the peptide abundance of auracyanin B increased only at the growing phase (Figure 3), suggesting that auracyanin B functions in photosynthetic electron transport, probably between the RC and ACIII.

Genes encoding respiratory-type ACIII-C_r_ subunits are clustered together with genes for cytochrome *c* oxidase (called COX or Complex IV) in a single operon in the genome of *C. aurantiacus* [26,34]. The respiratory-type ActBEAG (C_r_) peptides were detected during the lag phase and were less abundant during the growing phase (Figure 3). In a similar fashion, peptides derived from four subunits of cytochrome *c* oxidase were scarcely detected after 14 days (Figure 3). These results indicate the concerted expression of proteins for ACIII-C_r_ and terminal cytochrome *c* oxidase.

### 3.4. Central Carbon Metabolisms

*Chloroflexus* species use the oxidative tricarboxylic acid (TCA) cycle for NADH production and the 3-hydroxypropionate (3-OHP) bi-cycle for carbon fixation. The 3-OHP bi-cycle is found only in *Chloroflexota* bacteria [35,36]. Results of the proteomic analysis for proteins related to the central carbon metabolisms are shown as a pathway map in Figure 4. Most of the proteins related to the TCA cycle including glyoxylate bypass were abundant in the first lag phase (Figure 4, left-hand side). The result was consistent with the idea that the TCA cycle produces NADH for respiratory electron transfer under oxic conditions.

Two key enzymes for the 3-OHP cycle, malonyl-CoA reductase (*mcr*, Caur_2614) and propionyl-CoA synthase (*psc*, Caur_0613), were abundant in the growing phase (Figure 4). This clearly indicates that the 3-OHP pathway is active during phototrophic growth, although no CO_2_ was supplied to the culture medium.

The conversion of succinyl-CoA to malate is the common step in both the TCA cycle and the 3-OHP bi-cycle (Figure 4). Succinate is synthesized via two enzymatic reactions: CoA transfer from succinyl-CoA to malate by succinyl-CoA-L-malate CoA-transferase (Caur_0178–0179), and the reverse reaction by succinyl-CoA synthase (Caur_0702 and Caur_1121) with succinyl-CoA. Succinyl-CoA-L-malate CoA-transferase was abundant in the second growing phase, whereas succinyl-CoA synthase was abundant in the first lag phase. The synthesized succinate is oxidized by type B succinate dehydrogenase comprising an iron-sulfur subunit (*sdhB*, Caur_1880), a flavoprotein subunit (*sdhA*, Caur_1881), and a cytochrome *b* subunit (*sdhC*, Caur_1882) [37,38]. The expression patterns showed that SdhC was abundant in the first lag phase, whereas SdhAB was abundant in the second lag phase (Figure 4). Fumarate lyase (Caur_1443), which is involved in malate synthesis, was abundant in the first lag phase and in the phototrophic growing phase.

### 3.5. Pigment Compositions during the Cultivation

The pigment composition of *C. aurantiacus* cells was analyzed by reverse-phase HPLC. Two apparent peaks from carotenoids were detected at the retention times of 38.6 min and 42.8 min at days 0 to 10 (Figure 5A, peaks 1 and 2). These were temporarily assigned as 4-keto-γ-carotene and echinenone (ketolated β-carotene), based on their in-line absorption spectra (Appendix A). This is consistent with the protein expression data showing that carotenoid ketolase CrtO was abundant until day 10 (Figure 2). After day 12, two other peaks appeared at retention times of 49.2 min and 52.1 min, and they were assigned as γ-carotene and β-carotene, based on the previous reports (Figure 5A, peaks 3 and 4) [1,39]. These results indicate that carotenoid compositions were different from cells grown under oxic and anoxic conditions.

HPLC elution peaks from BChl *c* homologs were detected when measured at 667 nm (Figure 5B). Three peaks with retention times of 26.1, 31.6, and 36.1 were observed after day 6 (Figure 5B, peaks 5, 6, and 9). The in-line absorption spectra of the three elution peaks were identical to that of typical BChl *c* with Q_y_/Soret absorbance maxima at 667/434 nm (Appendix A). After day 20, two other small peaks could be seen at 34.7 and 35.2 min (Figure 5B, peaks 7 and 8). The in-line absorption spectrum of the small peak 8 showed Q_y_/Soret absorbance maxima at 657/427 nm (Appendix A), and therefore, the elution peak was ascribable to BChl *d*. Detection of the small amount of BChl *d* in *C. aurantiacus* was also reported previously [40].

Figure 5C shows the HPLC elution profile when absorbance was measured at 768 nm. Together with the elution peaks derived from BChl *c* (peaks 5, 6, and 9, as described above), a new substance was eluted at 33.7 min after day 12 (Figure 5C, peak 10). The in-line absorption spectrum of the peak 10 was typical to that of BChl *a* (Appendix A). These results indicate that BChl *a* started to be produced after day 12 in the growing phase, while BChl *c* already appeared after day 6 in the lag phase.

## 4. Discussion

*Chloroflexus aurantiacus* grows chemoheterotrophically under dark oxic conditions and photoheterotrophically under light anoxic conditions. Genes related to phototrophy are scattered in the genome of this organism [22]. This led to speculation that photosynthetic genes are not tightly regulated in *C. aurantiacus*. Proteomic analysis in this study revealed the expression dynamics of proteins involved in the photochemical RC complex, light-harvesting chlorosomes, pigment biosynthesis, electron transfer chains, and carbon fixation (Figure 2, Figure 3 and Figure 4). The results suggest the photosynthesis-related genes are concertedly regulated in *C. aurantiacus* despite the scattered distribution.

CsmMN reached almost the highest expression level at day 12 of cultivation (82–94% compared to the highest abundance at day 39, Figure 2), whereas CsmPOY increased gradually during the growing phase and reached their highest abundance at day 39. CsmMN proteins of *C. aurantiacus* are distantly related to CsmC/CsmD family envelope proteins of *C. tepidum*, which are involved in the assembly and glycolipid composition of the chlorosome envelope [41]. This suggests that CsmMN proteins, and probably also CsmA baseplate protein, are the components in the formation of premature, developing chlorosomes. On the other hand, CsmPOY could be important for the maturation of chlorosomes rather than for the initial structural formation.

BciC is a recently characterized enzyme for the removal of C-13^2^ methylcarboxyl groups which is the first committed step for BChl *c* biosynthesis [42]. Therefore, the enzymatic activity of BciC is essential for the formation of BChl *c* self-aggregates in chlorosomes. BciC reached its most abundant level at day 14 and then decreased to approx. one-fifth of its peak at day 17 (Figure 2). Simultaneously, the relative BChl *c* content in the cell did not increase after day 17 (Figure 1B). On the other hand, BchK was still abundant after day 17 of cultivation, and the highest abundance of BchK was at day 32. BchK is specific to BChl *c* biosynthesis and catalyzes esterification of the farnesyl tail at the final biosynthetic step of BChl *c*. The absorption peak of the monomer BChl *c* is at around 665 nm, while that of the aggregated BChl *c* is at around 740 nm. These results suggest that the self-aggregation of BChl *c* precedes the esterification of the hydrophobic farnesyl tail, and that BchK might work on the self-aggregated bacteriochlorophyllide *c*. *C. aurantiacus* is known to accumulate BChl *c* homologs with different alcohol chains at the carbon-17 position, usually geranylgeraniol, 9-octadecenol, and hexadecanol [43]. It was also reported that *C. aurantiacus* and *C. tepidum* can take up unusual alcohols for the long alkyl tail of BChl *c* when exogenous alcohols were supplied to the culture [40,44]. The low substrate specificity on the BChl tail esterification could be related to the speculated BchK enzymatic activity that the esterification occurred after BChl *c* self-aggregation.

There are two hypotheses for the process of the chlorosome biogenesis: “budding model” suggests the accumulation of BChl *c* and quinones in the cytoplasmic membrane triggers the chlorosome formation; “aggregation model” suggests CsmA-BChl *a* complexes are synthesized and assembled into a baseplate before the aggregation of other chlorosomal components and pigments (for the details of models, see references) [45,46,47]. The present study revealed that the BChl *c* production precedes the BChl *a* production in *C. aurantiacus* (Figure 2 and Figure 5). The results in this study support the “budding model”, at least in *C. aurantiacus*.

Mg-chelatase is composed of three subunits (BchH, BchI, and BchD). The *C. aurantiacus* genome contains three paralogs for *bchH* and two paralogs each for *bchI* and *bchD* genes. In *C. tepidum* in the phylum *Chlorobi*, three paralogs of *bchH* genes (CT1295, CT1955, and CT1957) were named *bchS*, *bchT*, and *bchH*, respectively. Two of the three *bchH* genes in *C. aurantiacus* (Caur_3151 and Caur_3371) are homologous to *bchH* of *C. tepidum*, and another (Caur_2591) is homologous to *bchS/T*, as determined by blastp analysis. Based on the protein expression patterns, all three homolog proteins of BchH seem to be active during the phototrophic growing phase. Two paralogs for *bchI* and *bchD* were likely to be expressed in different ways. *bchI*-1 (Caur_0419) and *bchD*-1 (Caur_0420) are included in the same operon as *bchF*, *bchC*, and *bchX*, and proteins from these genes were expressed during the growing phase (Figure 2). In contrast, BchI-2 (Caur_1255) and BchD-2 (Caur_0117) were expressed mainly during the lag phase, suggesting that these enzymes had different roles than in the photosynthetic process.

HPLC analysis demonstrated that carotenoid compositions were different between oxic and anoxic conditions (Figure 5). Two paralogs of CrtI showed opposite protein expression patterns: CrtI-1 (Caur_1413) was expressed in the lag phase and CrtI-2 (Caur_2422) was expressed in the growing phase (Figure 2). Together, these results imply that CrtI-1 and CrtI-2 are involved in the synthesis of ketolated and nonketo carotenoids, respectively.

Both ACIII-C_p_ and ACIII-C_r_ in *C. aurantiacus* are presumed to consist of the ActABCDEFG subunits [27]. However, the gene operon for the C_r_ type includes *actABEG* but does not include *actCDF*. In this study, peptides from ActCDF, encoded by genes in the C_p_ operon, were stably expressed in both the lag and growing phases, suggesting that the ActCDF subunits are used for not only C_p_ type but also C_r_ type ACIII supercomplex formation (Figure 6). Cytoplasmic ActABE subunits of C_p_ and C_r_ types are likely to be interchanged between oxic and anoxic conditions (Figure 6). The putative membrane-bound ActG subunit is a novel, specific protein for the *C. aurantiacus* ACIII [27] and is not present in ACIII of *R. marinus*.

The expression patterns of two auracyanins indicate that auracyanin A works with both ACIII-C_p_ and -C_r_, while auracyanin B works only with ACIII-C_p_ (Figure 3). During the phototrophic growing phase, the abundance of auracyanin B increased while that of auracyanin A decreased, suggesting that auracyanin B is more suitable than auracyanin A for photosynthetic electron transfer. *R. castenholzii* has a single auracyanin having auracyanin A-like N-terminal and auracyanin B-like C-terminal characteristics [48]. Proteome results showed that auracyanin B was specific to phototrophy in *C. aurantiacus*, implying that the C-terminal structure of *R. castenholzii* auracyanin could work for photosynthetic electron transport given that this auracyanin is important for both phototrophy and respiration.

Most of the enzymes involved in the 3-OHP bi-cycle were abundant during the first and second growing phases, indicating the involvement of the carbon fixation pathway for the phototrophic growth of *C. aurantiacus* under the tested conditions. Calvin cycle mutants of *Rhodospirillum rubrum* and *Rhodopseudomonas palustris* are reported to lose their ability to grow under anoxic light conditions due to electron imbalance rather than to the accumulation of toxic ribulose-1,5-bisphosphate [49]. In aerobic anoxygenic phototrophic bacteria, it is speculated that the lack of phototrophic growing ability under anoxic conditions is caused by the absence of RubisCO genes in the genome [50]. The active expression of 3-OHP enzymes after day 12 suggests the necessity of carbon fixation activity for the phototrophic growth of *C. aurantiacus*. In other words, the 3-OHP bi-cycle also may contribute to relaxing the electron imbalance during phototrophic growth.

## 5. Closing Remarks

The proteomic analysis in this study revealed the expression dynamics of a total of 2520 proteins out of 3934 CDSs deduced from the *C. aurantiacus* genome. Photosynthesis-related proteins were concertedly expressed during phototrophic growth, although the corresponding genes were scattered in the genome. The present results lead to the following hypothesis of how *C. aurantiacus* expresses proteins to change metabolic modes from respiration to phototrophy. For the structural formation of light-harvesting chlorosomes, the expression of CsmAMN proteins is important at the initial stage. BChl *c* was produced before BChl *a*, although the expression patterns of enzymes specific for BChl *c* and *a* biosynthesis are not very different. Formation of the BChl *c* self-aggregates could precede the esterification of hydrophobic tail by BchK, BChl *c* synthase. CrtI paralogs expressed in the lag phase and growing phase are involved in the synthesis of carotenoids with different chemical structures. Transmembrane proteins for ACIII are common for both ACIII-C_p_ and -C_r_, and relevant cytoplasmic subunits switch reversibly between respiratory and phototrophic conditions. Auracyanin A probably receives an electron from both ACIII-C_p_ and -C_r_ and transfers it to RC and cytochrome *c* oxidase, respectively. On the other hand, auracyanin B probably functions as the electron carrier between ACIII C_p_ and RC. The 3-OHP bi-cycle is important for not only photoautotrophic growth but also photoheterotrophic growth probably because it relaxes the electron imbalance. It is considered that functional genes for photosynthesis and carbon fixation in *Chloroflexota* members are obtained by horizontal gene transfers within the phylum and/or from other phyla [51,52,53,54]. Elucidation of the global regulatory system of genes involved in phototrophic growth will provide valuable insights into the identification of common features in nonphotosynthetic bacteria necessary for the acquisition of phototrophic ability through the horizontal gene transfer and will contribute to the understanding of the complex evolutionary process of phototrophic bacteria.

## Figures and Tables

**Figure 1 microorganisms-10-01288-f001:**
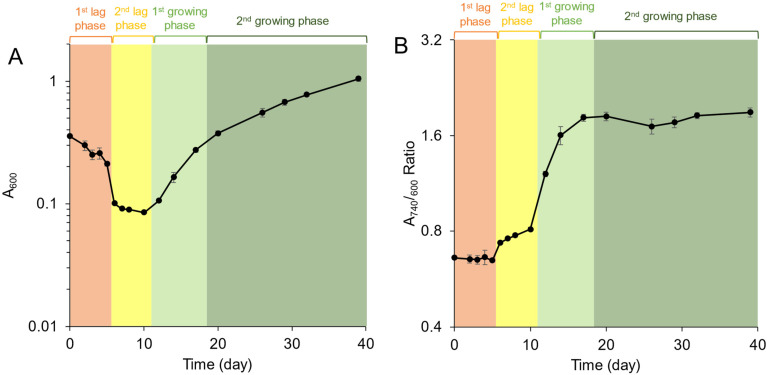
Photoheterotrophic growth profiles of *C. aurantiacus* represented by cell density values (**A**) and relative bacteriochlorophyll *c* contents (**B**). Measurements started after illumination. Orange, the first lag phase (days 0–5); yellow, the second lag phase (days 6–10); light green, the first growing phase (days 12–17); dark green, the second growing phase (days 20–39). The mean values of three measurements are shown with standard deviations.

**Figure 2 microorganisms-10-01288-f002:**
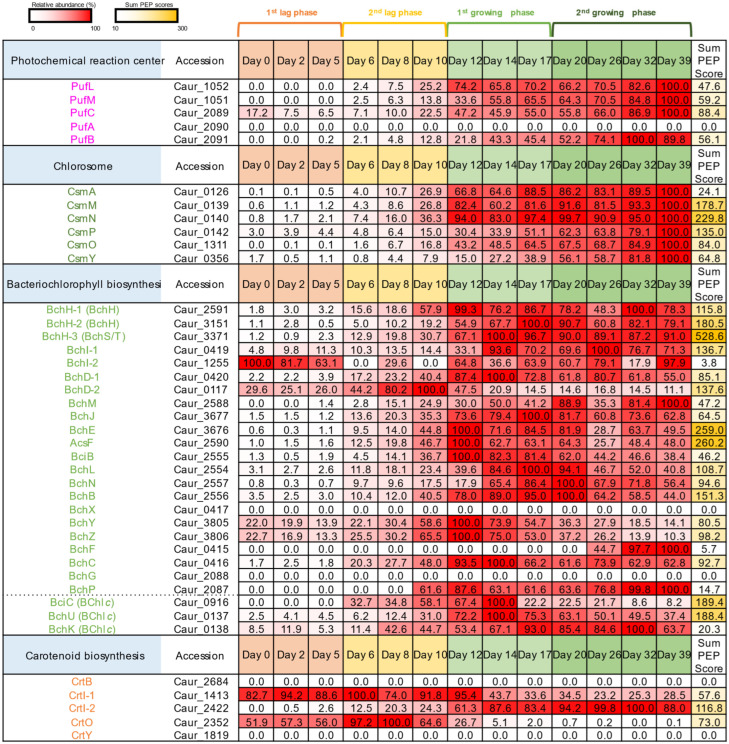
The relative abundance of proteins involved in reaction centers, chlorosomes, and the biosynthesis of bacteriochlorophylls and carotenoids. Relative values of protein expression among 13 sampling dates are shown as percentages indicating abundance, where “100%” indicates the highest protein expression among all sampling dates. The red color becomes deeper as the relative abundance values increase. Sum PEP scores are shown for the comparison of protein expression levels between different CDSs. The yellow color becomes deeper as the Sum PEP scores increase.

**Figure 3 microorganisms-10-01288-f003:**
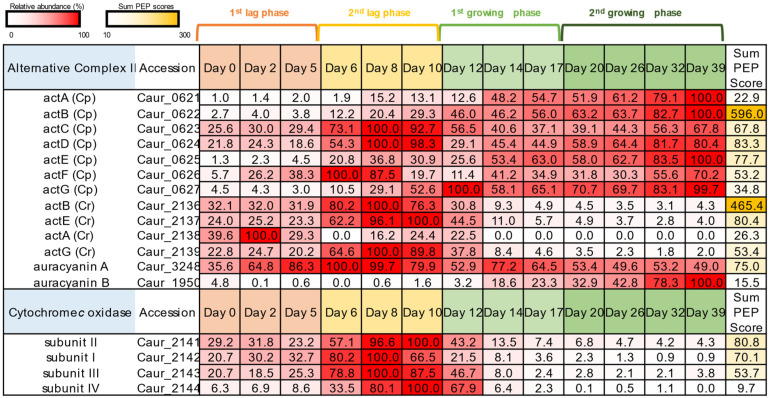
The relative abundance of proteins involved in photosynthetic and respiratory electron transfer chains. Relative values of protein expression among 13 sampling dates and Sum PEP scores are shown as described in Figure 2.

**Figure 4 microorganisms-10-01288-f004:**
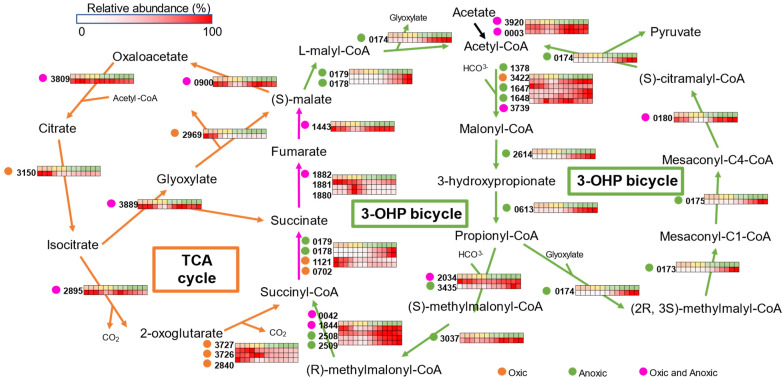
The heatmap showing the relative abundance of each protein involved in central carbon metabolisms in *C. aurantiacus*. The first row on the heat map corresponds to the number of cultivation days: orange, days 0–5; yellow, days 6–10; light green, days 12–17; dark green, days 20–39. Arrows indicate the enzymatic reactions for relevant metabolisms: orange, TCA cycle; green, 3-OHP bi-cycle; pink, both TCA and 3-OHP. Four-digit numbers on the left of heat maps represent locus tag numbers of *C. aurantiacus* genes.

**Figure 5 microorganisms-10-01288-f005:**
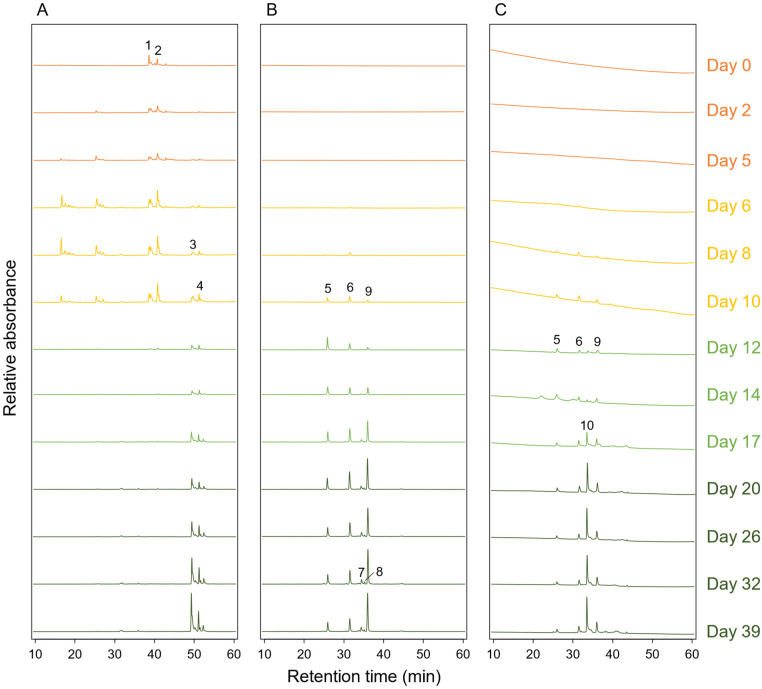
Reverse-phase HPLC elution profiles of pigments extracted from *C. aurantiacus* cells collected at the 13 sampling dates. Elution profiles when absorbance was measured at 490 nm, 667 nm, and 768 nm are shown in panels (**A**–**C**), respectively. The chromatograms in the first and second lag phases (days 0 to 10) were magnified five times for the clarity. The colors in elution profiles correspond to sampling dates: orange, days 0–5; yellow, days 6–10; light green, days 12–17; dark green, days 20–39. Putative assignments of peaks are as follows: peak 1, 4-keto-γ-carotene; peak 2, echinenone; peak 3, γ-carotene; peak 4, β-carotene; peaks 5, 6, 7, and 9, BChl *c*; peak 8, BChl *d*; peak 10, BChl *a*.

**Figure 6 microorganisms-10-01288-f006:**
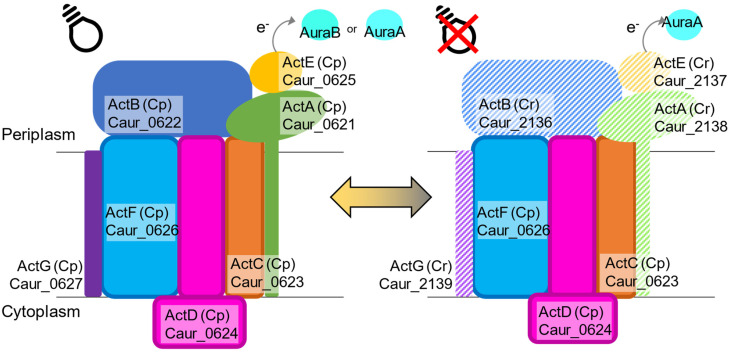
Schematic drawing of subunit compositions of alternative complexes III for photosynthesis (C_p_, **left**) and respiration (C_r_, **right**) in *C. aurantiacus*. The ActCDF subunits are commonly used in both ACIII-C_p_ and -C_r_. Two sets of ActABEG subunits are used for oxic dark and anoxic light conditions, respectively. Auracyanin A functions as the electron carrier with both C_p_- and C_r_-type ACIII, but auracyanin B works with the C_p_ type only.

## Data Availability

Not applicable.

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
