# Peer review of "Proteomic Time-Course Analysis of the Filamentous Anoxygenic Phototrophic Bacterium, Chloroflexus aurantiacus, during the Transition from Respiration to Phototrophy"

_microorganisms, 2022, doi:10.3390/microorganisms10071288_

Round 1
Reviewer 1 Report
The manuscript entitled "Proteomic time-course analysis of the filamentous anoxygenic phototrophic bacterium, Chloroflexus aurantiacus, during the transition from respiration to phototrophyprotein" dealt with proteomic time-course analysis of how C. aurantiacus expresses proteins to acclimate to the transition from chemoheterotrophic respiratory to photoheterotrophic growth mode. The paper is well written, but needs to be revised before publication.
According to the figure 1 legend “Orange, the first lag phase (days 0-5); yellow, the second lag phase (days 6-10); light green, the first growing phase (days 12-17); dark green, the second growing phase (days 20-39)”, several data are missing in Figure 1. Please correct it.
Figure 1A, in the Y-axis please indicate the cell density as number of cells/mL or gr of cells/mL.
Figure 1B, in the Y-axis please indicate BChl c content as mg/L or mg/mL.
Author Response
>The manuscript entitled "Proteomic time-course analysis of the filamentous anoxygenic phototrophic bacterium, Chloroflexus aurantiacus, during the transition from respiration to phototrophy" dealt with proteomic time-course analysis of how C. aurantiacus expresses proteins to acclimate to the transition from chemoheterotrophic respiratory to photoheterotrophic growth mode. The paper is well written, but needs to be revised before publication.
>According to the figure 1 legend “Orange, the first lag phase (days 0-5); yellow, the second lag phase (days 6-10); light green, the first growing phase (days 12-17); dark green, the second growing phase (days 20-39)”, several data are missing in Figure 1. Please correct it.
Thank you for your positive comments and also comments on Figure 1. The initially uploaded manuscript lacked background colors in Figure 1, probably because of the file conversion. We noticed it to the managing editor immediately after the initial submission and then uploaded the corrected version to the website, but it might not be in time. We apologize the confusion. This time, we upload the manuscript with the colored figure. In the newer version of Figure 1, we also added the statement of the growth phases on the top of the graphs, as shown in Figures 2 and 3.
>Figure 1A, in the Y-axis please indicate the cell density as number of cells/mL or gr of cells/mL.
>Figure 1B, in the Y-axis please indicate BChl c content as mg/L or mg/mL.
In Figure 1B, we would like to show changes of the relative amount of chlorosomes which contain BChl c self-aggregates (peaking at 740 nm), as a marker for the status that photosynthetic activities go on or off. Therefore, we did not utilize the value of BChl-c monomer amounts or concentrations. Please understand the situation. For the better comparison between Figures 1A and 1B, we would like to use A600 values for Figure 1A, too.

Reviewer 2 Report
The revised manuscript describes the study looking for the answer how Chloroflexus aurantiacus regulates protein expression in response to environmental changes. All studies providing insight into any processes/factors responsible for microorganisms' response and the change of their metabolism are important. For that reason I found the manuscript fitting into the scope of Microorganisms Journal.
The work is very well organized containing all required parts. Concise introduction provides key information on genes of interest. The aim and novelty is clear. Methods allow the repetition of the experiment. For me description is very good. The quality of data presentation is very good and it is easy to follow data description and presentation. They supplement discussion leading to concluding remarks.
In conclusion, the manuscript provides new knowledge about Chloroflexus aurantiacus metabolism switch after changing its growing condition. Proteomic studies allowed to identify proteins involved in the change.
Taking into account all above I recommend the manuscript for publication in the present form.
Author Response
>The revised manuscript describes the study looking for the answer how Chloroflexus aurantiacus regulates protein expression in response to environmental changes. All studies providing insight into any processes/factors responsible for microorganisms' response and the change of their metabolism are important. For that reason I found the manuscript fitting into the scope of Microorganisms Journal.
>The work is very well organized containing all required parts. Concise introduction provides key information on genes of interest. The aim and novelty is clear. Methods allow the repetition of the experiment. For me description is very good. The quality of data presentation is very good and it is easy to follow data description and presentation. They supplement discussion leading to concluding remarks.
>In conclusion, the manuscript provides new knowledge about Chloroflexus aurantiacus metabolism switch after changing its growing condition. Proteomic studies allowed to identify proteins involved in the change.
Taking into account all above I recommend the manuscript for publication in the present form.
Thank you for your positive comments. We are happy to hear that no revision is required.
Round 2
Reviewer 1 Report
The authors performed the requested changes and corrections. The paper is now suitable for publication.